# Melatonin Reverses the Warburg-Type Metabolism and Reduces Mitochondrial Membrane Potential of Ovarian Cancer Cells Independent of MT1 Receptor Activation

**DOI:** 10.3390/molecules27144350

**Published:** 2022-07-07

**Authors:** Maira Smaniotto Cucielo, Roberta Carvalho Cesário, Henrique Spaulonci Silveira, Letícia Barbosa Gaiotte, Sérgio Alexandre Alcantara dos Santos, Debora Aparecida Pires de Campos Zuccari, Fábio Rodrigues Ferreira Seiva, Russel J. Reiter, Luiz Gustavo de Almeida Chuffa

**Affiliations:** 1Department of Structural and Functional Biology, Institute of Biosciences, UNESP—São Paulo State University, Botucatu 18618-689, São Paulo, Brazil; maira.cucielo@gmail.com (M.S.C.); roberta.carvalho_@hotmail.com (R.C.C.); hspaulonci@gmail.com (H.S.S.); leticia.gaiotte@gmail.com (L.B.G.); sergio.santos@unesp.br (S.A.A.d.S.); 2Cancer Molecular Research Laboratory (LIMC), FAMERP, Department of Molecular Biology, São José do Rio Preto 15090-000, São Paulo, Brazil; debora.zuccari@famerp.br; 3Biological Science Center, Department of Biology, Luiz Meneghel Campus, Universidade Estadual do Norte do Paraná—UENP, Bandeirantes 86360-000, Paraná, Brazil; fabio.seiva@uenp.edu.br; 4Department of Cell Systems and Anatomy, UT Health, San Antonio, TX 78229, USA; reiter@uthscsa.edu

**Keywords:** ovarian cancer, melatonin, mitochondrial metabolism, glucose, Warburg effect, SKOV-3 cells

## Abstract

Ovarian cancer (OC) is the most lethal gynecologic malignancy, and melatonin has shown various antitumor properties. Herein, we investigated the influence of melatonin therapy on energy metabolism and mitochondrial integrity in SKOV-3 cells and tested whether its effects depended on MT1 receptor activation. SKOV-3 cells were exposed to different melatonin concentrations, and experimental groups were divided as to the presence of MT1 receptors (melatonin groups) or receptor absence by RNAi silencing (siRNA MT1+melatonin). Intracellular melatonin levels increased after treatment with melatonin independent of the MT1. The mitochondrial membrane potential of SKOV-3 cells decreased in the group treated with the highest melatonin concentration. Melatonin reduced cellular glucose consumption, while MT1 knockdown increased its consumption. Interconversion of lactate to pyruvate increased after treatment with melatonin and was remarkable in siRNA MT1 groups. Moreover, lactate dehydrogenase activity decreased with melatonin and increased after MT1 silencing at all concentrations. The UCSC XenaBrowser tool showed a positive correlation between the human *ASMTL* gene and the ATP synthase genes, succinate dehydrogenase gene (*SDHD*), and pyruvate dehydrogenase genes (*PDHA* and PDHB). We conclude that melatonin changes the glycolytic phenotype and mitochondrial integrity of SKOV-3 cells independent of the MT1 receptor, thus decreasing the survival advantage of OC cells.

## 1. Introduction

Ovarian cancer (OC) is the most lethal gynecologic malignancy and the fifth leading cause of death worldwide [1]. OC can originate from mesenchymal cells, granulosa cell layer, germinative cells, and epithelial cells, the last being responsible for about 90% of all OC types [2,3]. The standard treatment includes cisplatin–taxol-based chemotherapy; however, many tumors often relapse with aggressive features and become chemoresistant to conventional therapies [4,5].

Melatonin is secreted by the pineal gland and is produced, but not secreted, by extra-pineal tissues such as the retina, intestine, and ovary [6]. Melatonin’s actions are mediated by melatonin receptors (MT1 and MT2), or they are receptor-independent [7,8]. In addition to the regulatory role on circadian cycle, seasonal reproduction, immune response, etc. [9,10], evidence has shown its antitumor actions in several solid tumors such as prostate, breast, and colorectal cancers [11]. Its oncostatic effects include cell cycle arrest, proapoptotic, antiproliferative, and antimetastatic actions, in addition to interfering in mitochondrial physiology [12]. Some antitumor characteristics of melatonin are dependent on activation of the MT1 receptor; however, melatonin also exerts intracellular effects that are independent of MT1 [4]. Previous studies demonstrated that melatonin reaches specific subcellular compartments through the glucose transporter (GLUT1) and human peptide transporter 1 and 2 (PEPT1/2) [13,14,15].

Melatonin biosynthesis depends on biochemical processes such as the transformation of tryptophan into melatonin through the action of two limiting enzymes, arylalkylamine N-acetyltransferase (AANAT) and N-acetylserotonin-O-methyltransferase (ASMT) [16,17]. Both enzymes are present in mitochondria, strengthening the idea that melatonin is produced by perhaps all cells [17,18]. As a result, the maintenance of mitochondrial integrity is influenced by local melatonin production in healthy cells and, in tumor cells, mitochondrial dysfunction may be a result of altered melatonin synthesis that contributes to a defective cell phenotype [19].

Mitochondria are the organelles responsible for cell metabolic functions such as energy production, pyruvate metabolism, and cellular respiration (oxidative phosphorylation or OXPHOS) [15,20]. Metabolic reprogramming that induces high glycolytic rates even in the presence of oxygen (Warburg effect) is a hallmark of cancer [4,21]. In tumor cells, pyruvate is converted to lactate instead of mitochondrial acetyl-CoA, compromising the electron transport chain (ETC) and decreasing OXPHOS [22]. In addition, acetyl-CoA is a cofactor to the AANAT enzyme which is rate limiting in mitochondrial melatonin synthesis. The reduction in acetyl-CoA levels results in less melatonin for tumor cells and depresses the tricarboxylic acid cycle, which normally supports ATP production [15,23]. Against this background, the present study investigated the influence of melatonin therapy on energy metabolism and mitochondrial physiology in human ovarian carcinoma cells (SKOV-3 cell line) and further examined whether these biological effects are mediated by the MT1 melatonin receptor.

## 2. Results

### 2.1. Validation of the MT1 Receptor Silencing and Cytotoxicity Induced by Melatonin in SKOV-3 Cells

To better understand whether melatonin’s function in OC is dependent on the MT1 receptor, RNA interference (RNAi) was used to knockdown the melatonin receptor gene (MTNR1A and MTNR1B). After silencing, MTNR1A showed low relative gene expression (0.28 ± 0.10) compared to the negative silencing control group (si-NC) (1.01 ± 0.22) (~73% RNAi efficiency). The Western blot assay also showed a significant reduction in MT1 protein levels compared to the si-NC group. The expression of the MTNR1B receptor was not detected by RT–qPCR since it appeared to be very low or absent in the SKOV-3 cells (data not shown).

MTT assay was used to select a melatonin concentration with cytotoxic capacity and its further application in the cell culture. The melatonin concentrations showed ~20–45% cytotoxicity in SKOV-3 cells after 48 h. After melatonin treatment, SKOV-3 cells showed a statistically significant reduction in mitochondrial metabolism at all concentrations (1.6, 3.2, and 4 mM) (Figure 1). Interestingly, the siRNA MT1 + melatonin group exhibited a reduction in cell metabolism compared to the MT1 siRNA group and melatonin-exposed group at all concentrations (Figure 1A). Although melatonin significantly reduced cell viability, this effect was accentuated when MT1 was knocked down. Additionally, melatonin reduced cell size while increasing its granularity. Figure 1B shows the SKOV-3 cells after treatments in the presence or absence of MT1.

### 2.2. Melatonin Treatment Stimulates Intracellular Melatonin Synthesis

SKOV-3 cells showed a significant reduction in melatonin concentration compared with healthy ovarian cells (Figure 2A). Intracellular concentrations of melatonin increased in the OC cells exposed to 3.2 mM of melatonin compared to the control group (Figure 2B). Curiously, the siRNA MT1 + melatonin group showed a further increase in intracellular melatonin concentration compared to the control and melatonin-treated groups (Figure 2B). We used the GEPIA web-based tool to verify gene expression related to melatonin receptors and limiting enzymes of melatonin synthesis in OC patients. Although not statistically significant, the samples from OC patients had a higher expression rate of *MTNR1A* and *AANAT* genes but a reduced expression of *ASMT* gene compared with healthy ovaries (Figure 2C), thus indicating an impairment in melatonin synthesis by these pathological cells. The expression of the *MTNR1B* receptor was not observed in either tumor or normal tissue. Furthermore, there is a superior survival rate in patients who had a higher *ASMT* expression when compared with lower *ASMT* expression (Figure 2D).

### 2.3. Melatonin Alters the Mitochondrial Membrane Potential and Glycolytic Metabolism of SKOV-3 Cells

After melatonin exposure, MitoStatus Red labeling was performed to determine the mitochondrial membrane potential in SKOV-3 cells, in the presence or absence of an MT1 receptor, using flow cytometry. Mitochondrial membrane potential was lower after treatment with 4 mM of melatonin when compared to the control group. Moreover, MT1-silenced cells that received melatonin at concentrations of 3.2 and 4 mM exhibited a reduced membrane potential compared to the control, siRNA groups, and melatonin-treated groups, showing that melatonin exerts MT1-independent actions in the mitochondria (Figure 3A). Figure 3B shows the representative analysis of mitochondrial membrane potential in SKOV-3 cells.

The rate of cellular glucose consumption was reduced after exposure to 4 mM melatonin. The siRNA MT1+melatonin groups showed low rates of glucose consumption when compared with the control group. Furthermore, MT1-silenced SKOV-3 cells that received 1.6 and 3.2 mM of melatonin showed lower glucose consumption than their respective melatonin-treated groups (Figure 3C). On the contrary, glucose consumption in MT1-silenced SKOV-3 cells treated with 4 mM of melatonin was enhanced compared to the melatonin-treated group in the presence of MT1 (Figure 3C). Lactate dehydrogenase (LDH) activity, an enzyme that catalyzes the pyruvate to lactate interconversion, was reduced in SKOV-3 cells treated with 3.2 and 4 mM of melatonin compared to the control group. However, siRNA MT1 + melatonin groups showed an increase in LDH activity compared to control and melatonin-treated groups (Figure 3D). Analyzing the biological context, we observed that lactate was consumed rather than released to the extracellular medium. Thus, 3.2 mM of melatonin significantly reduced lactate levels during the treatment period compared to the control group. Interestingly, lactate consumption increased with MT1 knockdown considering all melatonin concentrations compared to the control and melatonin-treated groups (Figure 3E). It was suggested that MT1 receptors are essential to mediate glucose consumption in SKOV-3 cells treated with 4 mM melatonin showing an anti-Warburg effect in SKOV-3 cells. In the presence of MT1, the group treated with 3.2 mM of melatonin exhibited a reduction in lactate rates, but the knockdown of MT1 stimulated the oxidation of lactate to pyruvate by LDH as an alternative biochemical pathway. This enzyme showed a higher activity after MT1 knockdown compared to the melatonin-exposed group.

### 2.4. ASMTL Expression Is Positively Correlated with Specific Mitochondrial-Enzymes-Related Genes in OC

By using publicly available data in the TCGA platform, we performed a correlation analysis between the gene expression of the key limiting enzyme for melatonin synthesis, *ASMTL*, and the genes related to the molecules necessary for energy metabolism and mitochondrial respiration by comparing samples of OC patients and normal ovaries. Interestingly, *ASMTL*, which is a homologous gene that is highly expressed in OC, showed positive correlations with the pyruvate dehydrogenase E1 component subunit alpha and beta (*PDHA1*, *PDHB*) and succinate dehydrogenase (*SDHD*) genes and with ATP synthase subunits genes (*ATP5A1*, *ATPAF2*, *ATP5B*, *ATPAF1*, *C16orf7*, *ATP5C1*, *ATP5D*, and *ATP5E*) (Figure 4).

## 3. Discussion

Mitochondria are responsible for cellular bioenergetic regulation and respond to microenvironment changes [24]. It is well known that melatonin is capable of increasing apoptosis rates and promoting mitochondrial damage in tumor cells, mainly by stimulating ROS production and proapoptotic activity, in addition to reducing the mitochondrial membrane potential [14]. Melatonin treatment at all concentrations (1.6 to 4 mM) showed a marked cytotoxicity, being more pronounced at the highest concentration, consistent with the literature data [19,25,26,27]. Our present data revealed that melatonin application reduced the potential of the mitochondrial membrane and the size of SKOV-3 cells, and this process was independent of MT1 activation. The siRNA MT1+melatonin groups exhibited a decrease in mitochondrial membrane potential compared to the group that received melatonin at 3.2 and 4 mM concentrations. Under the pro-oxidant effect of melatonin in tumor cells, reactive oxygen species (ROS) may be produced and programmed cell death signaling initiated, thus causing damage to mitochondrial membrane integrity [28].

The presence of key enzymes for melatonin synthesis has been documented in many cell types other than those in the pineal gland so that tissues continuously express AANAT and ASMT regardless of the prevailing light:dark cycle [29]. In nontumor cells, these enzymes were found within the mitochondria to produce intracellular melatonin [18,30]. We observed a significant reduction in melatonin levels in OC cells compared to healthy ovarian cells; however, intracellular melatonin levels were restored after treatment with 3.2 mM regardless of the presence of the MT1 receptor. As already reported, higher intracellular melatonin concentrations may contribute to pro-oxidative, proapoptotic, and antiangiogenic processes and further change the metabolic profile of tumor cells [4,20]. We showed that melatonin treatment increased de novo intracellular melatonin concentration, and these findings seemed to be receptor-independent since the siRNA MT1 group actually exhibited higher intracellular melatonin levels compared to the control group. A plausible explanation for the low intracellular levels of melatonin in tumor cells is due to the limited availability of acetyl-CoA in pathological cells exhibiting Warburg-type metabolism; acetyl-CoA is a necessary cosubstrate for the limiting enzyme in melatonin synthesis, AANAT. In tumor cells, the pyruvate dehydrogenase complex (PDC) enzyme, which catalyzes the conversion of pyruvate to acetyl-CoA, is inhibited by the pyruvate dehydrogenase kinase (PDK) enzyme [16]. Melatonin may disinhibit PDC activity by reducing hypoxia-inducible factor 1-alpha (HIF-1α), a factor highly expressed in tumors due to hypoxic conditions, which stimulates PDK. After PDC disinhibition, melatonin is synthesized in mitochondria and OXPHOS is re-established [31].

Although tumor cells present high glycolytic rates, most cells are not capable of the noncytosolic lactate conversion cycle, which is known mainly as the aerobic glycolysis or the “Warburg effect” [32]. To assess this energy imbalance, we investigated the concentration of glucose, lactate, and LDH after exposure to melatonin, and the results showed that these molecules are susceptible to melatonin treatment. The rate of glucose consumption per time (Q) by SKOV-3 cells showed that 4 mM melatonin regulated glucose consumption differently in the experimental groups; there was a reduction in glucose uptake when compared to the control group in the presence of melatonin receptors. Conversely, the siRNA MT1 group treated with 4 mM of melatonin showed elevated glucose consumption; OC cells treated with 3.2 mM of melatonin showed higher lactate consumption per time (Q) compared to the control group. Cells with MT1 knockdown exhibited even higher lactate consumption at all melatonin concentrations compared to MT1 containing-cells treated with melatonin. Furthermore, melatonin treatment at concentrations of 3.2 and 4 mM reduced the LDH activity in SKOV-3 cells. Our data show that melatonin influences energy metabolism, with 3.2 and 4 mM melatonin being most effective. MT1 receptors seem to be related to glucose consumption, while the MT1-independent action may occur at the level of LDH activity by reverse metabolizing lactate to pyruvate in OC cells.

Treatment with melatonin alters the metabolic advantages of tumor cells by reversion of numerous tumor hallmarks associated with cell proliferation and survival. Previous studies indicate that, under specific pharmacological concentrations, melatonin affects cell proliferation by reducing glucose uptake [33,34,35,36], thereby altering the glucose metabolism of tumor cells. These studies were carried out using different cells types exposed to high melatonin concentrations (1 mM or even 5 to 10 mM) which are close to the concentrations used in our study. To ensure the oxidative profile of cells, optimal regulation of pyruvate metabolism is necessary by activating LDH [37,38]. The *LDH* gene encodes for two isoforms of the enzyme, LDHA and LDHB, the latter being able to convert lactate into pyruvate, making it available to enter the mitochondria and influence the TCA cycle [38,39]. The oxidation of lactate to pyruvate leads to an increase in protons, H+, acidification of lysosomes, and autophagy signaling by upregulation of sirtuin 5 (SIRT5); as a result, there is lactate uptake and cytotoxicity effects, as shown in the MTT assay [40]. This could explain the highly elevated LDH activity and lactate consumption in the MT1 silenced cells. These data show that melatonin acts via the MT1 receptor, decreasing glucose consumption and reversing the Warburg phenotype. In the absence of MT1, however, melatonin utilizes other possible mechanisms, possibly oxidizing lactate to pyruvate resulting in an anti-Warburg effect associated with a poorer cell fate. In addition, in the mitochondria, melatonin inhibits the PDK enzyme, which allows the conversion of pyruvate into acetyl-CoA, a component of the TCA cycle, causing a possible restoration of the normal metabolic phenotype of cells that use OXPHOS for energy generation [20]. Melatonin can also modulate several mitochondrial pathways in SKOV-3 cells and redirect cells to the oxidative mitochondrial phenotype, leading to a reversal of the malignant phenotype of cells [21,41]. These receptor-independent findings might be associated with other transport routes, such as involving GLUT1 and PEPT1/2.

Correlation analysis performed using GEPIA showed differences in the gene expression of AANAT and ASMT between OC samples and nontumor tissue. The expression of these enzymes-encoding genes occurs naturally in the ovaries for melatonin production to maintain tissue integrity by scavenging ROS release during ovulation [42]. We observed a decrease in the expression of ASMT in OC, corroborating the present findings where tumor tissue had a lower concentration of melatonin. The ASMT enzyme exhibits an important dimorphism related to melatonin synthesis, such as ASMTL, being essential for females, which are assumed to have a higher reserve capacity for melatonin production than males [17]. When analyzing the survival curve of patients with high and low expression of *ASMT*, we observed a decrease, by several months, in the survival rate of patients with lower *ASMT* expression. Tran et al. (2021) analyzed breast cancer patients, and those with increased ASMT levels showed longer survival rate and were metastasis-free [43]. These data support that tumor cells with gene mutations that depress melatonin synthesis have advantages for tumor progression.

Additional investigation of *ASMTL* expression correlated with particular genes related to TCA cycle and the ETC. We observed that *ASMTL* showed positive correlation with *PDHA* and *PDHB*, *SDHD*, and the ATP synthase-related genes. Downregulation of PDC enzymes, including PDHA and PDHB causes a slowdown of OXPHOS rate while increasing the glycolysis rate to obtain ATP. In turn, melatonin favors the formation of acetyl-CoA from pyruvate due to the PDC disinhibition, with acetyl-CoA being an essential intermediate for the TCA cycle and a cofactor for the AANAT enzyme. TCA plays a central role in the catabolism of carbohydrates, fatty acids, and amino acids via aerobic respiration. The SDHD, an enzyme belonging to the mitochondrial complex II, plays an essential role in the ETC and the TCA cycle. Treatment of uterine endometrial cancer cells with melatonin halts tumor progression through SDH inhibition [44]. Moreover, we previously reported that female reproductive organ tumors exhibit a positive correlation of *ASMT* with OXPHOS-related enzymes including *PDHA1* and *SDHB* [45]. Mitochondria have the machinery responsible for the conversion of redox energy into an electrochemical gradient and the production of ATP formed in ETC through ATP synthase [46]. ATP synthase has a structure divided into two fractions, F0 in the mitochondrial membrane, and F1 in the mitochondrial matrix. In addition, each fraction has subunits transcribed by different genes and functionalities such as catalytic or motor [47]. Through the proteomic profile, Chuffa et al. [48] showed overexpression of the beta subunit ATP synthase after melatonin treatment in a rat model of ovarian cancer. The ability of melatonin to alter the characteristics of the glycolytic metabolism of tumor cells has been recently reported, especially by upregulating proteins related to TCA cycle and ETC [41]. Our findings document a positive correlation between the enzyme-related genes responsible for melatonin synthesis and the alpha and beta subunits, corroborating other previous findings. Figure 5 summarizes the main cellular mechanisms after melatonin treatment in SKOV-3 cells.

Although there is a sequential relationship between the glycolytic pathway and mitochondrial metabolism, the Warburg effect is not necessarily linked to disruptions on that transition step. In tumor cells, the Warburg effect can be observed even in the presence of intact mitochondria with oxidative phosphorylation events and ATP production. We speculate that the effect of melatonin on energy metabolism may be more related to gene modulation or to changes in the concentrations of reactants and products in the reaction catalyzed by LDH. Quantification of the activity of specific enzymes related to the glycolytic pathway, and the investigation of the intracellular concentrations of glucose, lactate, and ATP, will help to obtain more detailed responses; to solve these limitations, ongoing studies by our group have these purposes.

## 4. Material and Methods

### 4.1. Cell Line and Cell Culture

SKOV-3 cell line was purchased from the American Type Culture Collection (ATCC, Rockville, MD, USA) and cultured in RPMI 1640 (LGC Biotechnology, Cotia, SP, Brazil) supplemented with 10% fetal bovine serum (FBS; Gibco, Waltham, MA, USA), 100 U/mL penicillin and 100 μg/mL streptomycin (Gibco, Waltham, MA, USA). The cells were maintained in a humidified atmosphere with 5% CO_2_ at 37 °C.

### 4.2. Melatonin Preparation

Melatonin (Sigma-Aldrich, St. Louis, MO, USA) was dissolved in dimethyl sulfoxide (DMSO, Merck, Germany) to prepare a 1 M stock solution. Then, different concentrations of working solution were prepared in RPMI 1640 medium and incubated with SKOV-3 cells.

### 4.3. Experimental Groups

SKOV-3 cells (1 × 10^5^ cells) were seeded and incubated with melatonin (1.6, 3.2, and 4.0 mM concentrations) or not (control group), with or without siRNA for MT1 receptor (20 nM). In individual sets of experiments, SKOV-3 cells were incubated with melatonin in the presence of MT1 receptors (melatonin-exposed groups) or after MT1 silencing (siRNA MT1+melatonin-exposed groups) for 48 h. The control groups either received 1% DMSO (control group) or were silenced for MT1 and received no treatment (siRNA MT1). All the experiments were assayed in three technical and biological replicates.

### 4.4. Oligonucleotides and Transfection

Post-transcriptional silencing of the MT1 gene was performed using RNA interference (RNAi). Two Silencer^®^ Select siRNA sequences (s224070 and s9051, Thermo Fisher, Waltham, MA, USA) formed a complex with Opti-MEM^®^ Reduced Serum Medium (Thermo Fisher, Waltham, MA, USA) before transfection. SKOV-3 transfections were performed with RNAiMAX Lipofectamine (Thermo Fisher, Waltham, MA, USA) combined with 20 nM of each oligonucleotide for 24 h when the cells reached 80% of confluence. The respective negative control was used (Silencer^®^ Select Negative Control No. 1 siRNA, Thermo Fisher, Waltham, MA, USA). The time and concentration of the oligonucleotides were previously determined in a pilot study. The silencing validation was done through RT–qPCR.

### 4.5. RNA Isolation, RT–qPCR, and Western Blotting

After transfection, cells were washed with D-PBS, and the RNA isolation was performed using TRIzol ^®^. RNA concentration and quality were assessed using a NanoVue Plus Spectrophotometer (GE Healthcare, Wauwatosa, WI, USA). To validate MT1 receptor silencing, reverse transcription–quantitative PCR (RT–qPCR) was used. Total RNA samples were reverse transcribed into cDNA using the High Capacity RNA-to-cDNA Kit (Life Technologies). mRNAs qPCR analysis was performed in 20 μL reaction (SYBR Green Master Mix; Thermo Fisher Scientific, Waltham, MA, USA), according to the manufacturer’s instruction, and run on a QuantStudio 12K Flex System (Thermo Fisher Scientific, Waltham, MA, USA), using the following conditions: 95 °C for 10 min followed by 40 cycles of 95 °C for 15 s and 60 °C for 1 min. The oligonucleotides for mRNAs were analyzed through the National Center for Biotechnology Information (NCBI) and the sequences used: MTNR1A (MT1) 5′-AGCTCAGGAACGCAGGAAAC-3′ (forward) and 5′-CAGTGCAGATAGCCCAGGTT-3′ (reverse); RPS13 5′-AGAAACGGCTACCACATCCA-3′ (forward) and 5′-CACCAGACTTGCCCTCCA-3′ (reverse); GAPDH 5′-GCTCCCTCTTTCTTTGCAGCAAT-3′ (forward) and 5′-TACCATGAGTCCTTCCACGATAC-3′ (reverse). The RT–qPCR data were presented as fold-change (2^−ΔΔCt^) compared with the endogenous genes (RPS13 and GAPDH). Total protein was extracted using RIPA lysis buffer, followed by quantification using a NanoVue^®^ spectrophotometer (GE Healthcare) and run on 10% SDS-PAGE. MT1 primary antibody (ab116337, abcam, 1:500 dilution) and secondary antibody (1:10,000 dilution) were used to verify the bands after adding the ECL^®^Selected Western Blotting Detection Reagent detection system (GE Healthcare, Uppsala, Sweden).

### 4.6. Cytotoxicity Assay

The SKOV-3 cells were seeded in a 96-well plate at a density of 1 × 10^4^ cells/well. The cell survival was measured using a colorimetric method. After melatonin exposure, the MTT solution (5 mg/mL) was added for 4 h, and the formazan crystals were dissolved using DMSO. The concentration was determined by an Epoch microplate reader (BioTek Instruments, Highland Park, PO, USA) at 540 nm, being the reference curve fixed at 650 nm. The percentage of crystal formation was calculated by fixing the control group as 100%.

### 4.7. Melatonin Quantification by ELISA Assay

Intracellular melatonin concentration was measured using the Human Melatonin ELISA Kit (FineTest, catalog number EH3344). To compare melatonin concentration in normal and tumor cells, we used transformed ovarian cancer cells and cells from normal ovarian tissue. The protein concentration used in all samples was 3 µg. For this purpose, a 96-well plate coated with antibody to melatonin received 50 µL of sample or standard solution per well, and then 50 µL of biotin was added followed by incubation for 45 min at 37 °C. Subsequently, the plate was washed, and 100 µL of HRP-streptavidin solution was added for 30 min at 37 °C. After this procedure, the plate was washed and incubated with 90 µL of TMB solution for 20 min at 37 °C, and finally, 50 µL of stop solution was added and read at 450 nm in a microplate reader (Epoch, BioTek Instruments, Santa Clara, CA, USA). All analyses were performed in triplicate, and the standard curve was generated according to the manufacturer’s instructions. Data were expressed in pg/mL.

### 4.8. Metabolic Status of the OC Cells

To evaluate the effect of melatonin on the energy metabolism of SKOV-3 cells, biochemical assays were used to detect lactate release, glucose consumption, and LDH activity. Cells were seeded in 12-well plates for glucose and lactate analysis (5 × 10^4^ cells/well) and in six-well plates for LDH analysis (1 × 10^5^ cells/well). After reaching 80% of cell confluence, the supernatant was collected for analysis (t0), the cells were trypsinized, and viable cells were counted (×0). Next, the cells with or without the MT1 receptor were exposed to melatonin for 48 h. After the exposure period, the supernatant was collected for analysis (t1) and the cells were trypsinized, with only the viable cells counted (×1). The supernatants were analyzed using a Bioclin Quibasa Química Básica Ltd. enzyme kit (Belo Horizonte, MG, Brazil). Specific events involved in the Warburg effect, including glucose consumption, lactate release, and LDH activity, were analyzed by enzymatic assay after 48 h of melatonin exposure. The specific rate of substrate consumption and metabolite production was calculated using the formula Q = 2 × 10^−3^ (C_t0_ − C_t1_)/[(X_t1_ + X_t0_) × t]; Q is the specific consumption rate (mmol cell h^−1^); C refers to metabolite concentration (mmol cell^−1^ h^−1^); t_0_ and t_1_ are the plated seeded times of cells (t_0_ = 0 h) and supernatant collection (t_1_ = 48 h); X_t0_ and X_t1_ are the number of viable cells at both times (cell mL^−1^) [49]. The experiments were performed in biological triplicate.

### 4.9. Target Prediction and Bioinformatics Analyses

The GEPIA 2 database (http://gepia.cancer-pku.cn/, accessed on 18 March 2022) provides an essential interactive function including differential expression analysis, correlation analysis, and patient survival analysis [50]. We used this web-based tool to compare the gene expression of melatonin receptors (MT1 and MT2) and the genes of the limiting enzymes for melatonin synthesis (AANAT and ASMT) in normal ovarian tissue (88 samples) and ovarian cancer (426 samples).

We also analyzed the correlation between differential gene expression of *ASMTL* with ATP-synthase-related genes (*ATP5A1*, *ATPAF2*, *ATP5B*, *ATPAF1*, *C16orf7*, *ATP5C1*, *ATP5D*, and *ATP5E*); pyruvate dehydrogenase genes (*PDHA1* and *PDHB*); and succinate dehydrogenase complex gene (*SDHD*) available from the Cancer Genome Atlas (TCGA; ovarian cancer dataset (n = 634 samples)) using the UCSC Xena browser (http://xenabrowser.net/, accessed on 18 March 2022) [51]. 

### 4.10. Measurement of Mitochondrial Membrane Potential 

Flow cytometry was used to determine the mitochondrial membrane potential integrity using MitoStatus Red Kit (BD Pharmingen™, San Diego, CA, catalog number 564697). According to the manufacturer’s recommendations, cells were incubated with 50 nM of MitoStatus Red at 37 °C for 30 min and then washed using D-PBS. The assay was performed using the FACSCanto cytometer (BD Biosciences, Clontech, CA, USA). The relative rates of cells with low membrane potential were calculated using FlowJo software (vX.10.6, Tree Star Inc., Ashland, Vilmington, DE, USA).

### 4.11. Statistical Analysis

Data were evaluated using analysis of variance (ANOVA) with independent factors complemented with Tukey’s test for multiple comparisons. For nonparametric data, the Kruskal–Wallis test was used, complemented with Dunn’s test. To assess the effects of MT1 silencing, we applied the Student’s *t*-test between the silencing negative control group (si-NC) and the MT1-silenced group (siRNA MT1). Results were analyzed using GraphPad Prism software version 9.2, and data were expressed as mean ± SD. Statistical significance was set at *p* < 0.05.

## 5. Conclusions

In summary, we demonstrated that melatonin is an essential antitumor agent and possesses a critical role in energy metabolism and mitochondrial integrity. At the highest concentrations, melatonin reversed glucose uptake through the MT1 receptor in SKOV-3 cells. In addition, an increase in lactate consumption was apparent possibly due to the LDH reverse conversion of lactate to pyruvate in a MT1-independent manner. Additional studies will provide further details on the role of melatonin in the signaling pathways that control cellular energy generation and metabolism. OC cells usually present lower enzyme levels for melatonin synthesis, and treatment with melatonin augments its intracellular concentration. Moreover, our results reinforce the evidence that melatonin has both MT1-dependent and MT1-independent anticancer effects in OC.

## Figures and Tables

**Figure 1 molecules-27-04350-f001:**
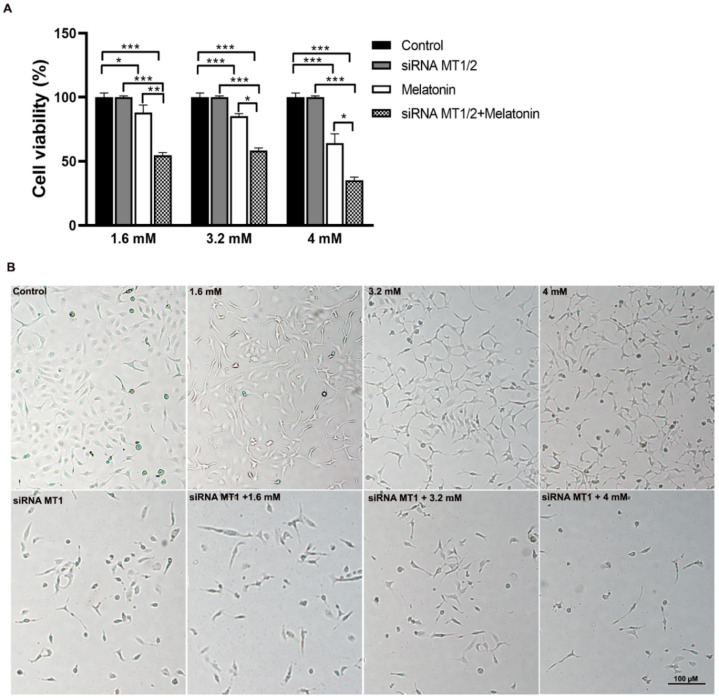
Melatonin decreased the mitochondrial activity. (**A**) MTT assay showed that cytotoxicity was increased after melatonin treatment with different concentrations for 48 h. (**B**) Representative images of the SKOV-3 cells after melatonin treatment in the presence or absence of MT1 receptors; the assay were obtained at 40× magnification The samples were assayed in three technical and biological replicates. The data are expressed as the mean ± SD. * *p* < 0.05, ** *p* < 0.01, and *** *p* < 0.001.

**Figure 2 molecules-27-04350-f002:**
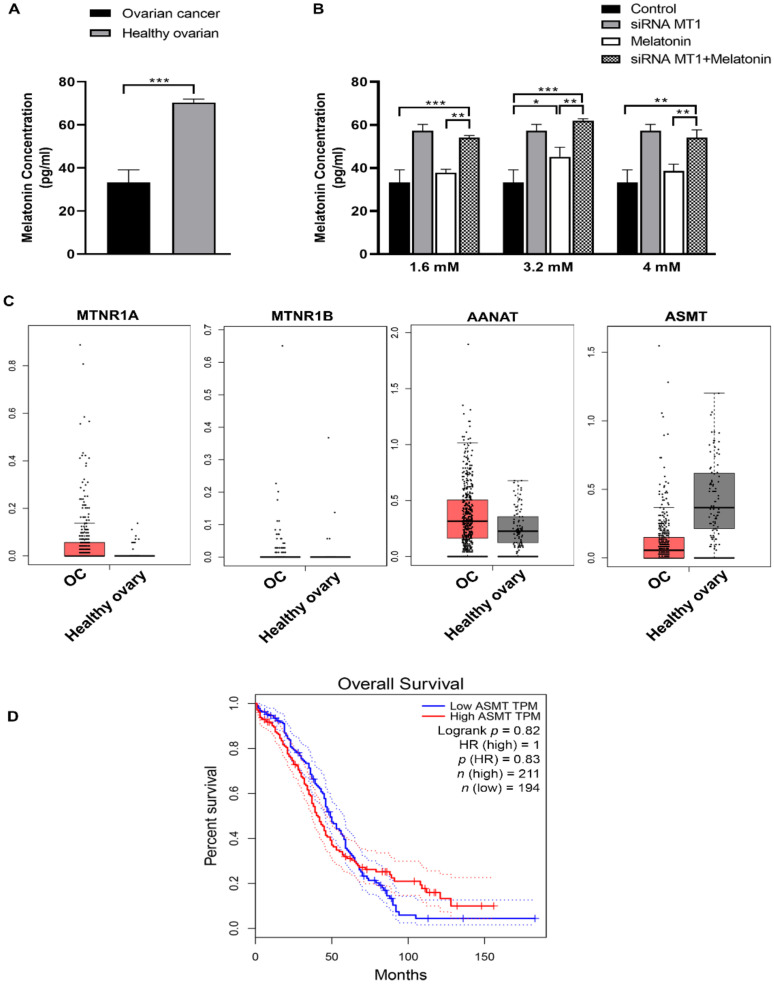
Intracellular melatonin concentration. (**A**) The melatonin concentration in SKOV-3 cells and healthy ovarian tissue. (**B**) Intracellular melatonin concentration in SKOV-3 cells, in the presence of MT1 receptor or not after melatonin for 48 h. (**C**) Expression levels of MT1 and MT2 melatonin receptors, and the limiting enzymes of melatonin synthesis (AANAT and ASMT) in patients with OC (red box) and healthy tissue (gray box) from the TCGA datasets. (**D**) The overall survival of OC patients with low *ASMT* (blue line) and high *ASMT* (red line). Images were obtained from the GEPIA online database. These analyses were made using different experimental conditions and their respective control groups. Data are expressed as the mean ± SD. * *p* < 0.05, ** *p* < 0.01, *** *p* < 0.001.

**Figure 3 molecules-27-04350-f003:**
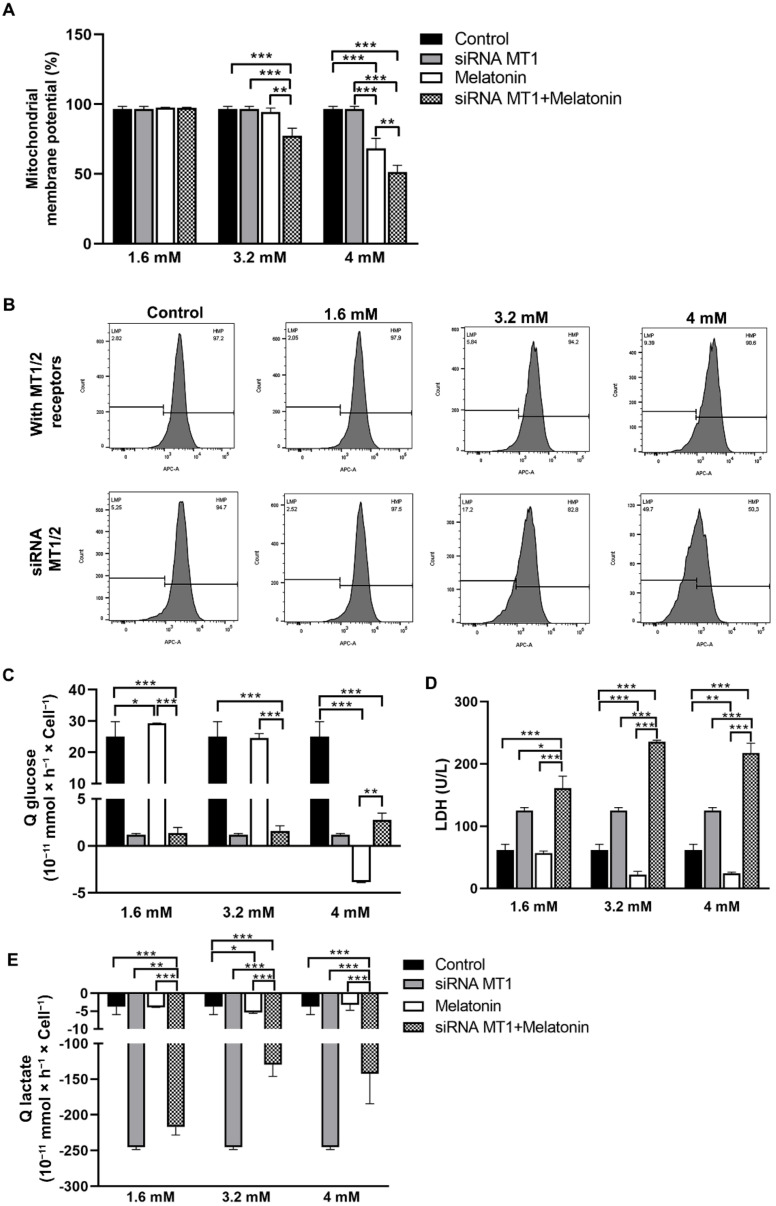
Melatonin alters mitochondrial integrity and reduces the Warburg effect in SKOV-3 cells. *(***A**) Mitochondrial membrane potential rate in SKOV-3 cells, in the presence or absence of MT1, after melatonin treatment for 48 h. MitoStatus Red was used to detect cells by flow cytometry. (**B**) Representative analysis of mitochondrial membrane potential in SKOV-3 cells. (**C**) Glucose consumption. (**D**) LDH activity. (**E**) Lactate consumption in SKOV-3 cells after melatonin treatment, in the presence or absence of MT1. The analysis was made using different experimental conditions and the respective control groups. Data are expressed as the mean ± SD. * *p* < 0.05, ** *p* < 0.01, *** *p* < 0.001.

**Figure 4 molecules-27-04350-f004:**
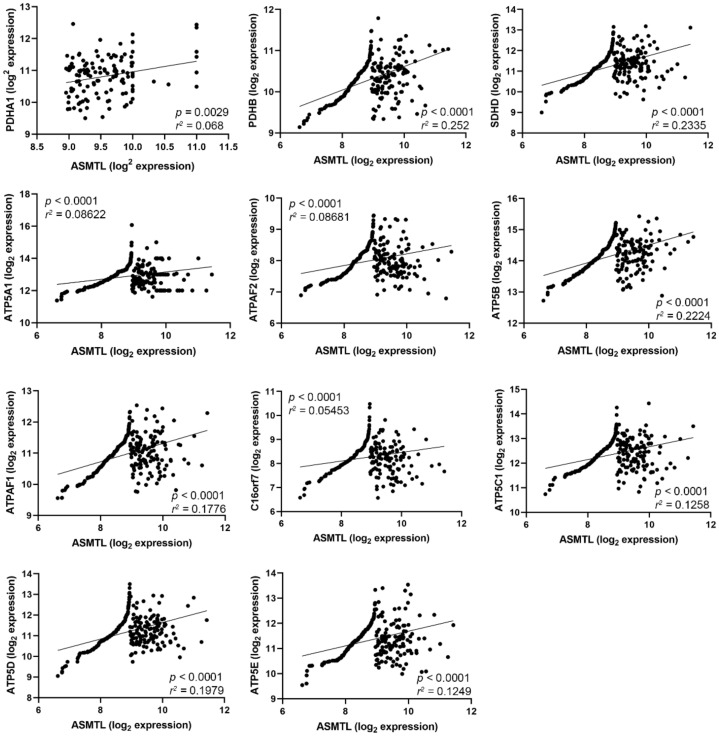
Correlation between *ASMTL* gene expression and the genes associated with mitochondrial energy metabolism and ATP production in human OC. Pearson’s correlation coefficient was used individually for each comparison. UCSC Xena Browser was used to identify the relative gene expression profile in OC samples available from the TCGA database.

**Figure 5 molecules-27-04350-f005:**
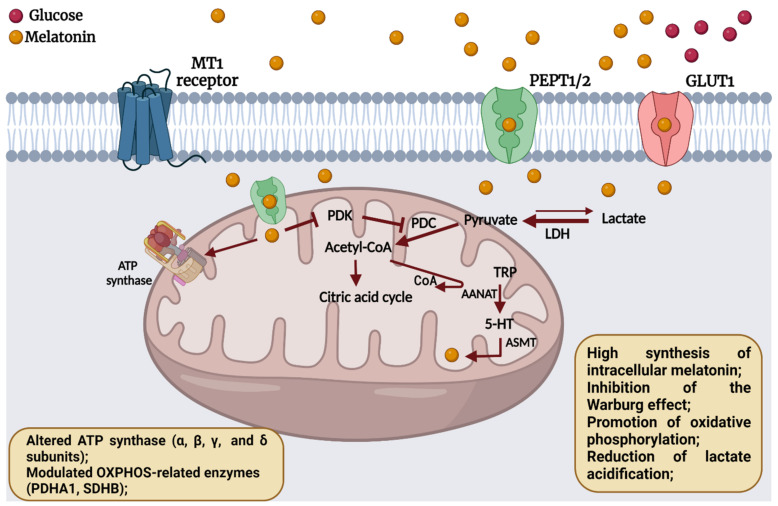
The diagrammatic representation shows potential mechanisms modulated by melatonin. Melatonin inhibits pyruvate dehydrogenase kinase (PDK), an enzyme that suppresses the dehydrogenase complex, responsible for converting pyruvate into acetyl-Coenzyme A (acetyl-CoA). The concentration of pyruvate can be increased by the enzyme lactate dehydrogenase (LDH) reverse conversion from lactate. The increased availability of pyruvate for conversion to acetyl-CoA for the citric acid cycle leads to the reversal of the Warburg effect; acetyl-CoA also is a limiting factor for the functioning of the AANAT enzyme, an intermediary in the synthesis of intramitochondrial melatonin. Furthermore, melatonin alters the function of ATP synthase, compromising the integrity of the mitochondrial membrane and inducing cell death. TRP: tryptophan; 5-HT: 5-hydroxytryptophan. Arrow: activation; bar-headed lines: inhibition. Created with BioRender.com (2022).

## Data Availability

Not applicable.

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
