# Peer review of "Melatonin Reverses the Warburg-Type Metabolism and Reduces Mitochondrial Membrane Potential of Ovarian Cancer Cells Independent of MT1 Receptor Activation"

_molecules, 2022, doi:10.3390/molecules27144350_

Round 1
Reviewer 1 Report
The present draft addresses an interesting and pertinent topic. Cucielo and colleagues analysed the capacity of melatonin to reverse the Warburg-type metabolism in ovarian cancer cells.
Although this is a very interesting hot topic, some experiments are needed. For instance, the authors use the MTT assay to analyze cell viability. However, the MTT assay analyses mitochondrial metabolic activity, not cell viability. In this context, other assays should be performed.
Consistently with these observations, I believe this manuscript could be considered for publication only if it is improved, and if the following points are properly approached:
- The MTT assay analyzes mitochondrial metabolic activity. Since melatonin probably interferes with mitochondrial activity, another assay should have been chosen, as the trypan blue exclusion method. I recommend performing a specific assay to quantify cell viability.
- In order to better understand the metabolic/bioenergetic alterations, it would be important to quantify ATP content. Please elaborate on this.
- The assessment of glycolytic metabolism will benefit from the quantification of intracellular lactate and glucose levels. Please elaborate on this.
- What is the association between the concentrations used in the study and the pharmacological concentrations mentioned in the Discussion section? Please elaborate on this.
- All abbreviations should be given in full the first time they are mentioned within the text.
- Please revise formatting, as recommended by the Molecules Journal.
- In the Material and Methods section: please correct “1x105 cells” to “1x105 cells”. Please revise all text.
- In the Material and Methods section: please specify the DMSO concentration used in the control group.
- In the Material and Methods section: please specify the RT-qPCR complete program.
- In the Material and Methods section: please correct “2-ΔΔCt” to “2-ΔΔCt”
- In the Material and Methods section: MT1 protein levels were analysed through a Western Blotting assay. Please provide details on this procedure.
- In the Material and Methods section: were the formazan crystals dissolved?
Author Response
Reviewer #1
The present draft addresses an interesting and pertinent topic. Cucielo and colleagues analysed the capacity of melatonin to reverse the Warburg-type metabolism in ovarian cancer cells. Although this is a very interesting hot topic, some experiments are needed. For instance, the authors use the MTT assay to analyze cell viability. However, the MTT assay analyses mitochondrial metabolic activity, not cell viability. In this context, other assays should be performed.
Consistently with these observations, I believe this manuscript could be considered for publication only if it is improved.
Response: We really appreciate all of the points/issues raised by the reviewer in order to improve the quality and scientific value of our manuscript. We did our best to address all raised issues.
- The MTT assay analyzes mitochondrial metabolic activity. Since melatonin probably interferes with mitochondrial activity, another assay should have been chosen, as the trypan blue exclusion method. I recommend performing a specific assay to quantify cell viability.
Response: We agree with the reviewer. We added the images of cytotoxicity to demonstrated cell amount and morphology to rule out a possible direct effect of melatonin on mitochondria. We choose to use the term "cytotoxicity" instead of "cell viability" which cause too divergences among the researchers (please see highlighted text on page 3 and Figure 1 B).
- In order to better understand the metabolic/bioenergetic alterations, it would be important to quantify ATP content. The assessment of glycolytic metabolism will benefit from the quantification of intracellular lactate and glucose levels. Please elaborate on this.
Response: This point is very consistent. For these criticisms, we include in the text a topic regarding the limitations of our study, and briefly discuss them in a few details (please see discussion section on pages 11-12). We are conducting a new research with focus on the metabolic activity of specific enzymes associated with aerobic glycolysis and, in that study, we will also evaluate intracelular levels of lactate, glucose, and glutaminolysis.
- What is the association between the concentrations used in the study and the pharmacological concentrations mentioned in the Discussion section? Please elaborate on this.
Response: Thank you for considering this important issue. The pharmacological concentration of melatonin used in in vitro studies concerning the glucose metabolism is rather variable but usually in order of mM. A new sentence has been added for a better contextualization of our study (please see highlighted text on page 10).
- All abbreviations should be given in full the first time they are mentioned within the text.
Response: Thank you for advising us. We have checked all the abbreviations to be sure the full name was added when first mentioned in the text.
- In the Material and Methods section: please correct “1x105 cells” to “1x105 cells”. Please revise all text.
Response: Thanks for raising this point. This has been changed as requested.
- In the Material and Methods section: please specify the DMSO concentration used in the control group.
Response: Thanks for raising this point. This has been specified as requested.
- In the Material and Methods section: please specify the RT-qPCR complete program.
Response: Thanks for raising this point. This has been changed as requested.
- In the Material and Methods section: please correct “2-ΔΔCt” to “2-ΔΔCt”
Response: Thanks for raising this point. This has been changed as requested.
- In the Material and Methods section: MT1 protein levels were analysed through a Western Blotting assay. Please provide details on this procedure.
Response: Thanks for raising this point. Details of the procedure has been added to materials and methods.
- In the Material and Methods section: were the formazan crystals dissolved?
Response: Thanks for raising this point. Yes, the formazan crystals were dissolved.
We have complemented this sentence.
Reviewer 2 Report
The present study investigates the influence of melatonin on the metabolism (intracellular melatonin synthesis, mitochondrial membrane potential, glycolytic metabolism, ASMTL expression) of SKOV-3 cells and melatonin receptor knock down regulated SKOV-3 cells. The authors demonstrate in the concentration range of 1.6 to 4mM melatonin (371 - 928 mg/l). Inhibition of cell growth in non-knock down regulated cells, an increase in melatonin concentration in ovarian carcinoma cells, reduction of mitochondrial membrane potential, reduced glucose consumption and lower LDH activity in non-down regulated cells and a positive correlation of ASMTL expression with genes important for ATP synthesis.
As a result of their investigations, the authors design a reaction scheme that promotes the conversion of pyruvate by pyruvate dehydrogenase into acetyl CoA by inhibiting pyruvate dehydrogenase kinase.
The experiments are clearly carried out and described. The study should be published in the present form.
Author Response
Reviewer #2
The present study investigates the influence of melatonin on the metabolism (intracellular melatonin synthesis, mitochondrial membrane potential, glycolytic metabolism, ASMTL expression) of SKOV-3 cells and melatonin receptor knock down regulated SKOV-3 cells. The authors demonstrate in the concentration range of 1.6 to 4mM melatonin (371 - 928 mg/l). Inhibition of cell growth in non-knock down regulated cells, an increase in melatonin concentration in ovarian carcinoma cells, reduction of mitochondrial membrane potential, reduced glucose consumption and lower LDH activity in non-down regulated cells and a positive correlation of ASMTL expression with genes important for ATP synthesis.
As a result of their investigations, the authors design a reaction scheme that promotes the conversion of pyruvate by pyruvate dehydrogenase into acetyl CoA by inhibiting pyruvate dehydrogenase kinase.
The experiments are clearly carried out and described. The study should be published in the present form.
Response: We really appreciate the recommendation of our study as it follows. The metabolic activities of the specific enzymes associated with aerobic glycolysis are being evaluated in another research.
Round 2
Reviewer 1 Report
The correction and clarification of points improved the quality of the manuscript. I believe that this manuscript version should be considered for publication.